# Relationship between scapular initial position and scapular movement during dynamic motions

Jun Umehara[1,2]*, Masahide Yagi[1], Tetsuya Hirono[1,2], Tomohito Komamura[3], Satoru Nishishita[1], Noriaki Ichihashi[1]

**1** Human Health Sciences, Graduate School of Medicine, Kyoto University, Kyoto, Japan, **2** Research Fellow of Japan Society for the Promotion of Science, Tokyo, Japan, **3** Chiba University Hospital, Rehabilitation unit, Chiba, Japan

* umehara.jun.77z@st.kyoto-u.ac.jp

**Data Availability Statement:** All relevant data are within the manuscript.

**Funding:** This work was supported by a grant-in-aid from the Japan Society for the Promotion of

## Abstract

Optimal scapular position and movement are necessary for normal function of the shoulder joint and it is essential to focus on scapula in the rehabilitation for shoulder disorders. The aim of this study was to discover the relationship between the scapular initial position and scapular movement during dynamic motions in healthy young men. Thirty-four men participated in this study. The scapular angles at initial position and in elevation and lowering during flexion and abduction were measured using an electromagnetic tracking device. The scapular movements from 30˚ to 120˚ during flexion and abduction were calculated. Spearman's rank correlation coefficients were used to analyze the relationship between the scapular initial position and scapular movements. For upward rotation and posterior tilt of the scapula, there were significant positive correlations between the scapular initial position and scapular movement during flexion and abduction. For internal rotation, there were significant positive correlations, except 90˚ in lowering phase and 120˚ in both phases. While the humeral elevation increased, the correlation coefficients tended to decrease. Except for the internal rotation our results clarified the interactions between the scapular initial position and scapular movement during dynamic motions in healthy young men. The tendency of the decrease in correlation coefficient with elevation angle was shown.

## Introduction

The shoulder complex consists of the glenohumeral-, acromioclavicular-, and scapulothoracic-joints and has the largest range of motion in the body[1]. Normal shoulder function needs optimal scapular position and its movement because the scapula has an integrable role for the shoulder complex. Abnormal motion and position of the scapula are defined as scapular dyskinesis[2] and present as shoulder disorders[3–6]. Preventive strategies, treatments, and development of clinical tests for shoulder rehabilitation, therefore, should focus on scapular position and movement.

Science for Young Scientists (18J12658). The funder had no role in study design, data collection and analysis, decision to publish, or preparation of the manuscript.

The scapular initial position is probably defined by the shape of the thorax, passive tension of the scapulothoracic muscle, and acromioclavicular articulation. Previous studies investigating the scapular initial position showed upward rotation of the scapula of 5.4˚; downward rotation of 2˚; internal rotation of 26.5˚, 40˚, and 41.1˚; and anterior tilt of 2˚ and 13.5˚[7–11]. In perspective, the scapula at initial position is positioned approximately in the middle of the upward downward rotation, 35˚ internal rotation, and 10˚ anterior tilt[11].

For the scapular movement, the scapulothoracic muscles, which include the trapezius-, serratus anterior-, pectoralis minor-, levator scapula-, and rhomboid-muscles[12], control scapular movement during elevation. In particular, the trapezius and serratus anterior muscles work in coordination as a coupled force, which is needed for optimal scapular movement[13]. Upward and internal rotation and posterior tilt of the scapula generally occur with humeral elevation in healthy people[8,14,15].

Sahrman[16] described that normal alignment at the static position (i.e. resting position) is of need for normal joint movement and asserted the importance of the relationship between alignment and joint movement. In the clinical setting, Reijneveld et al.[17] and Strufy et al.[18] advocated that the assessment of scapular position and scapular-conscious exercise were implemented to improve scapular movement during humeral elevation as part of shoulder rehabilitation. They also suggested the importance of the relationship between alignment and joint movement. One previous study, to our knowledge, developed a thorax-fixed regression model for prediction of the scapular orientation from the humeral orientation using static position of upper limb (i.e. non-dynamic motion) and fixed-thorax posture [19], which may differ from actual dynamic shoulder motion. Therefore, no studies have focused on the interaction between scapular initial position and movement during dynamic motion, although scapular initial position and its movement during elevation and lowering has been frequently measured in shoulder biomechanics research. Understanding of interaction between scapular initial position and its consequential movement is meaningful for both therapeutic rehabilitation and biomechanics research. The aim of this study was to clarify the relationship between scapular initial position and scapular movement during dynamic motions in healthy young men. We hypothesized that there is a positive correlation between the two variables in healthy young men.

## Materials and methods

### Participants

Thirty-four men [mean age, 22.7 (3.1) years; mean height, 170.8 (5.4) cm; mean weight, 65.6 (8.1) kg] were recruited at our university and participated in this study. The upper limb used to throw a ball was defined as the dominant limb. Participants currently with or a history of a neurological or orthopaedic disorder in their nondominant limb were excluded. Given the measurement error, we also excluded women because they would have more subcutaneous fat than men [20], which could affect to the sensor, particularly the thoracic sensor. At recruitment, three men (one with shoulder impingement syndrome and two with a history of clavicle fracture) were excluded. The aim and procedure of the study were provided to all participants, who then provided informed consent. The study protocol was approved by the ethics committee of Kyoto University Graduate School and the Faculty of Medicine (R0233) and conformed to the principles of the Declaration of Helsinki.

### Experimental procedures

All procedures of this study were conducted in our laboratory. To measure the scapular initial position, participants were asked to sit on a stool in a relaxed manner with their upper limbs

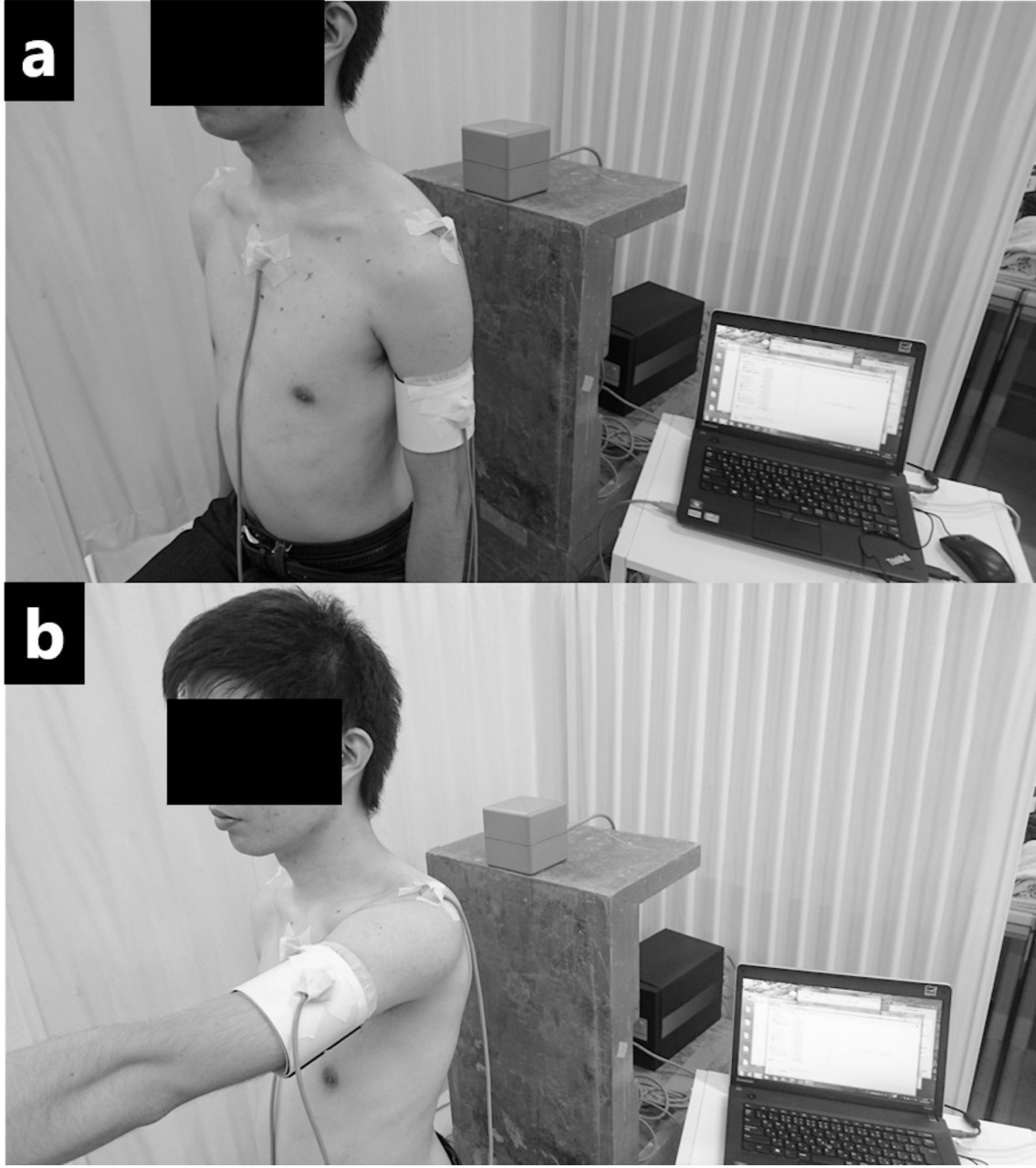

**Fig 1. Experimental procedures.** The scapular initial position and movement were assessed dung sitting (a) and dynamic motions (b) respectively. The individual in this manuscript has given written informed consent (as outlined in PLOS consent form) to publish these case details.

and the palms beside their body. No instructions were given on how to position the lower limbs, pelvis, trunk, and head so each participant could reflect his natural posture ([Fig 1A]). To measure scapular movement during dynamic motions i.e. elevation and lowering, the participants sitting on the stool were asked to raise and lower their nondominant arm in the sagittal (flexion) and frontal (abduction) planes. From the starting position, with the upper limb beside their body, the elbow fully extended, and the palm against the body, the participants fully elevated their upper limb in 4 s and then lowered it to the starting position in 4 s three

consecutive times with the use of a metronome at 60 beats/min (Fig 1B). The participants fixed their eyes on a target placed in front of them at eye level. The scapular initial position was measured first followed by random measurements of scapular movements. The participants underwent sufficient familiarization sessions before measurements began.

## Instrumentation

The angles of the scapula and the humerus in three dimensions were measured at rest and during flexion and abduction using a 6-degrees-of-freedom electromagnetic tracker (Liberty, Polhemus, Colchester, VT, USA) at 120 Hz on the nondominant limb. This system comprises a stylus, five sensors, and a transmitter controlled by an electronic unit. The accuracy of the sensor is 0.762 mm in terms of position and 0.15 degrees in terms of orientation. The transmitter was put on a wooden stand 30 cm behind the participants and at height of 100 cm. The transmitter emitted an electromagnetic field that detected the sensors and the stylus. The global coordinate system was used to represent the electromagnetic field, with the X-, Y-, and Z-axes being the forward, upward, and right directions, respectively, and the transmitter was the origin. The sensors were placed on the skin over the bony landmarks using adhesive tape. The thoracic sensor was attached to the sternum inferior to the jugular notch, the humeral sensor was attached to the midpoint of the humerus using a thermoplastic cuff, and the scapular sensor was attached to the plateau of the acromion. The location of these sensors formed the local coordinate system of the thoracic, humeral, and scapular segment respectively through the digitization of each bony landmark. These procedures were implemented on all participants while they sat on the stool.

## Data processing

All local coordinate systems were according to the shoulder standardization proposal of the International Society of Biomechanics [21]. The rotations were represented by the distal coordinate system relative to the proximal coordinate system using the Euler angle (Fig 2). Details is written in our previous study[22]. All data were processed using MATLAB software (MathWorks, Natick, MA, USA).

The scapular initial position was measured for 3 s while the participants sat on the stool; the mean value was calculated and used for further analysis. The scapular movements during flexion and abduction were measured while the humeral angle ranged from 30˚ to 120˚ relative to the thorax. The data from the three movements was averaged. The mean value every 30˚ was then calculated and used for further analysis. In addition, integral values and the coefficient of variance (CV) of scapular movement were calculated as the humerus was elevated from rest to 120˚ in 10˚ increments, which means scapular kinematic characteristic with respect to humeral movement and its' variability among participants. The humeral angle upto 120˚ was chosen for scapular movement analysis because of the small effect of soft tissue on the surface measurement [23], and because the data could be collected for all participants. The intraclass correlation coefficient (1,3) calculated from the three scapular movements in ten healthy men [mean age, 25.1 (3.2) years; mean height, 171.1 (6.7) cm; mean weight, 60.5 (13.8) kg] is shown in Table 1, with a high enough reliability.

## Statistics analysis

All data were analysed using SPSS Statistics 22 software (IBM, Armonk, NY, USA). The Shapiro Wilk test was used to confirm the normality of the data; it showed non-normal distributions in some of the data. The Friedman test was used for each scapular rotation (i.e., internal external rotation, downward upward rotation, and anterior posterior tilt) to investigate

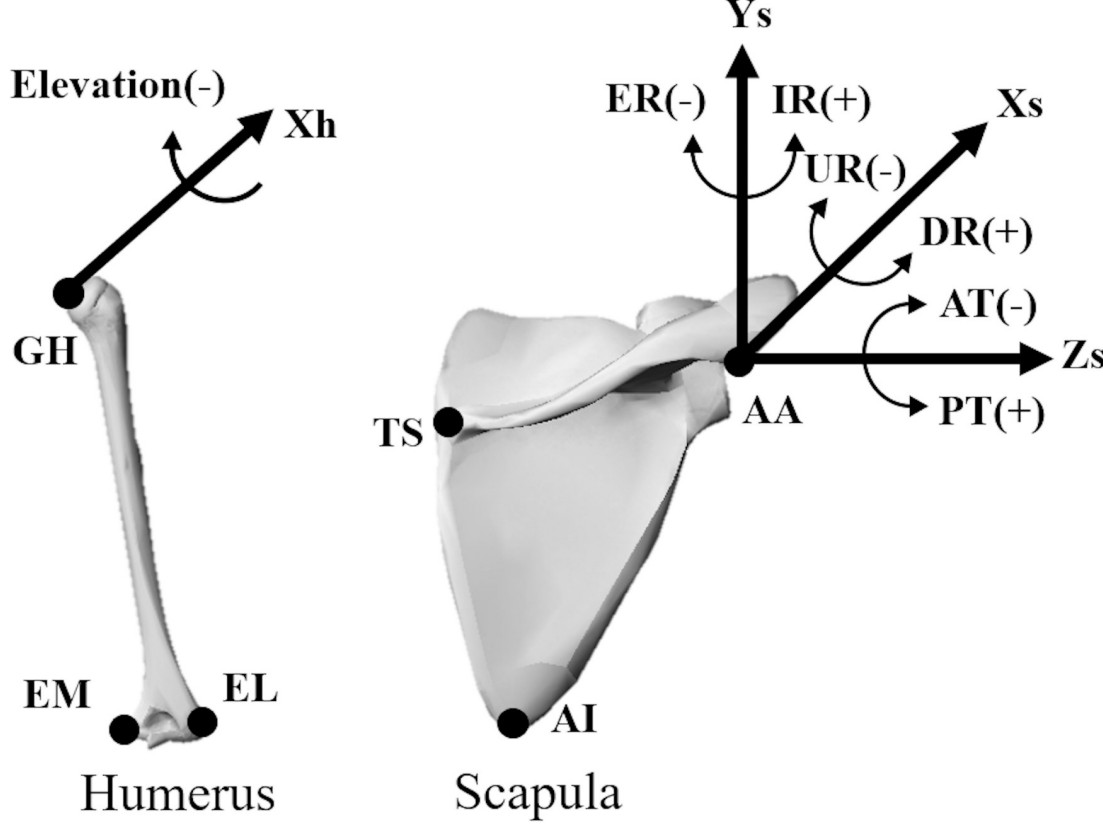

**Fig 2. Definition of the coordinate system.** In the local coordinate system of the humerus (left), the Xh-axis was perpendicular to the plane defined by the glenohumeral rotation centre (GH), lateral epicondyle (EL), and medial epicondyle (EM). In the local coordinate system of the scapula (right), the Xs-axis was perpendicular to the plane defined by the trigonum spina scapula (TS), acromial angle (AA), and inferior angle (AI). The Ys-axis was defined as the cross product of the Xs-axis and the Zs-axis. The Zs-axis was defined as the direction from the TS to the AA. ER, external rotation; IR, internal rotation; UR, upward rotation; DR, downward rotation; AT, anterior tilt; PT, posterior tilt.

**Table 1. Measurement reliability for scapular movements.**

| | | Elevation phase | | | | Lowering phase | | | |
|---|---|---|---|---|---|---|---|---|---|
| | Elevation angle | 30° | 60° | 90° | 120° | 120° | 90° | 60° | 30° |
| Flexion | Internal/External rotation | 0.98 (0.96–0.99) | 0.98 (0.95–0.99) | 0.97 (0.91–0.99) | 0.94 (0.83–0.98) | 0.97 (0.92–0.99) | 0.98 (0.94–0.99) | 0.98 (0.95–0.99) | 0.98 (0.94–0.99) |
| | Downward/Upward rotation | 0.98 (0.96–0.99) | 0.98 (0.94–0.99) | 0.95 (0.86–0.98) | 0.98 (0.95–0.99) | 0.99 (0.97–0.99) | 0.98 (0.94–0.99) | 0.94 (0.82–0.98) | 0.86 (0.61–0.96) |
| | Posterior/Anterior tilt | 0.97 (0.93–0.99) | 0.98 (0.94–0.99) | 0.98 (0.96–0.99) | 0.98 (0.94–0.99) | 0.98 (0.95–0.99) | 0.97 (0.93–0.99) | 0.96 (0.88–0.98) | 0.91 (0.75–0.97) |
| Abduction | Internal/External rotation | 0.94 (0.83–0.98) | 0.97 (0.93–0.99) | 0.99 (0.98–0.99) | 0.99 (0.97–0.99) | 0.99 (0.97–0.99) | 0.99 (0.98–0.99) | 0.98 (0.96–0.99) | 0.98 (0.95–0.99) |
| | Downward/Upward rotation | 0.97 (0.92–0.99) | 0.99 (0.97–0.99) | 0.99 (0.98–0.99) | 0.99 (0.98–0.99) | 0.99 (0.98–0.99) | 0.99 (0.98–0.99) | 0.97 (0.92–0.99) | 0.97 (0.93–0.99) |
| | Posterior/Anterior tilt | 0.99 (0.97–0.99) | 0.99 (0.97–0.99) | 0.98 (0.96–0.99) | 0.99 (0.97–0.99) | 0.99 (0.97–0.99) | 0.99 (0.98–0.99) | 0.97 (0.93–0.99) | 0.99 (0.97–0.99) |

The values are intraclass correlation coefficients (1,3) and the range in parentheses is the 95% confidence interval.

whether the scapular initial position differed from scapular movement during flexion and abduction. When a significant main effect was confirmed, the Wilcoxon signed rank test with Bonferroni correction for post hoc analysis was performed to compare the scapular initial position with scapular movement at each humeral elevation. Spearman's rank correlation coefficient was calculated to examine the relationship between the scapular initial position and scapular movement at each humeral angle for each scapular rotation. A confidence level of 0.05 was used in all of the statistical tests.

## Results

All data were represented by a median value (25%, 75%) because some data had a non-normal distribution. The scapular initial position had an internal rotation of 29.7° (24.0°, 32.9°), upward rotation of 1.6° (3.4°, 4.1°), and posterior tilt of 5.2° (6.9°, 3.7°). The Friedman test indicated main effects of scapular rotation variable. Then post hoc test showed the significant differences between the scapular initial position and almost scapular movements in each rotation. No significant differences were found between the scapular initial position and the posterior tilt at 30° in lowering phase during flexion and in both phases during abduction (Figs 3 and 4). The integrated amount of change in the scapular movement and the CVs for every 30° are presented in Table 2. CVs between participants tended to be small for the upward rotation and large for the internal rotation and posterior tilt.

There were significant positive correlations between the scapular initial position and scapular movements with respect to upward rotation and posterior tilt at all humeral angles in elevation and lowering phase during flexion. For internal rotation, the scapular position in elevation phase showed significant positive correlation with scapular movements at humeral elevations of 30°, 60°, and 90° even though significant positive correlations between two variables at all humeral angles were found in lowering phase during flexion. As well as the correlations during flexion, there were significant positive correlations between the scapular initial position and scapular movements with respect to upward rotation and posterior tilt at all humeral angles during abduction. For internal rotation, the scapular position showed significant positive correlations with scapular movements at all humeral elevations except 90° in lowering phase and 120° in both phases. In addition, correlation coefficients for each scapular rotation during flexion and abduction with humeral elevation tended to be small (Table 3).

## Discussion

Our study investigated the relationship between the scapular initial position and scapular movement in elevation and lowering phase during flexion and abduction and showed that there were significant positive correlations between these variables except that at a few humeral angles in healthy young men. In addition, these correlation coefficients tended to decrease with humeral angle during flexion and abduction. To the best of our knowledge, this is the first study to focus the interaction between scapular initial position and its movement and to demonstrate significant correlations between them in elevation and lowering phase during flexion and abduction, i.e. dynamic motions, in healthy young men. Our results partly supported our hypothesis that the scapular initial position is related to scapular movement during dynamic motions in healthy young men.

In the present study, the three-dimensional scapular initial position represented about 30° of internal rotation, the middle of the upward/downward rotation, and about 5° of anterior tilt. The scapula rotated upward, externally, and then internally and tilted posteriorly during flexion, and rotated upward and externally and tilted posteriorly during abduction. Strufy et al.[11] demonstrated that the scapula at rest (i.e. initial position) was approximately

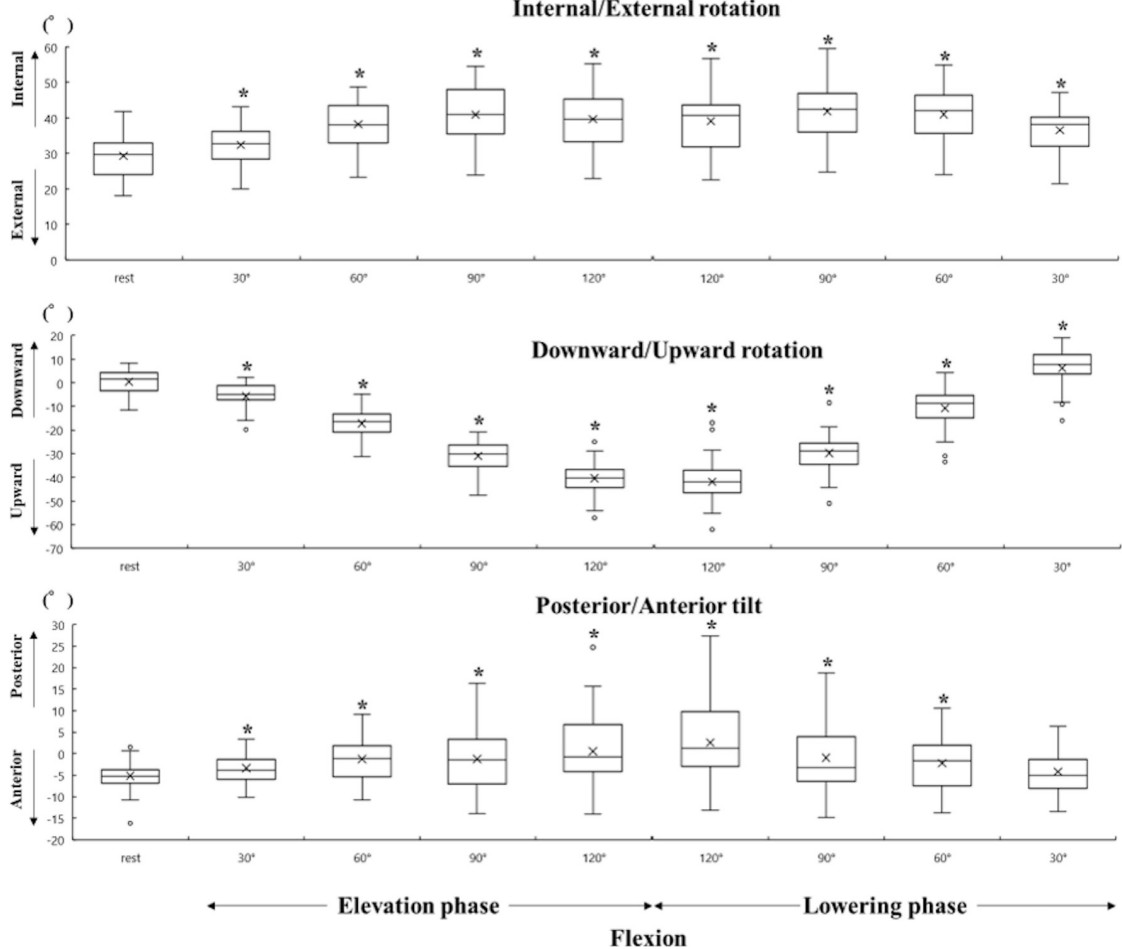

**Fig 3. Scapular position and movement during flexion.** Box-and-whisker plots show internal and external rotation (top), downward and upward rotation (middle), and anterior and posterior tilt (bottom) of the scapula. The X-axis is the humeral angle and the Y-axis is the scapular angle. The asterisk represents the significant difference compared with the scapular position at rest (P < 0.05).

horizontal, had an internal rotation of 35˚, and an anterior tilt of 10˚ with some extent individual difference. Previous studies that investigated the three-dimensional scapular angle during arm elevation reported that the scapula had upward rotation, posterior tilt, and internal rotation, followed by external rotation during flexion, whereas external rotation occurred throughout the abduction phase[7,8,15]. Our study therefore most likely is in same lines of abovementioned ones.

Our results clarified the significant positive correlations between the two variables in scapular downward upward rotation and anterior posterior tilt, except internal external rotation, during flexion and abduction. To our knowledge, there is one study to directly explore the relationship between the scapular initial position and the consequently scapular movement. de Groot and Brand[19] developed a thorax-fixed regression model for prediction of the scapular orientation from the humeral orientation, and they showed that the scapular orientations were predicted by humeral variables, external road, and scapular initial position. Our study could be the same line of this previous study even though the direct comparison between two studies is complicated because their experimental procedure constituted of static position of upper limb (i.e. non-dynamic motion) and fixed-thorax posture, which may be far from actual dynamic

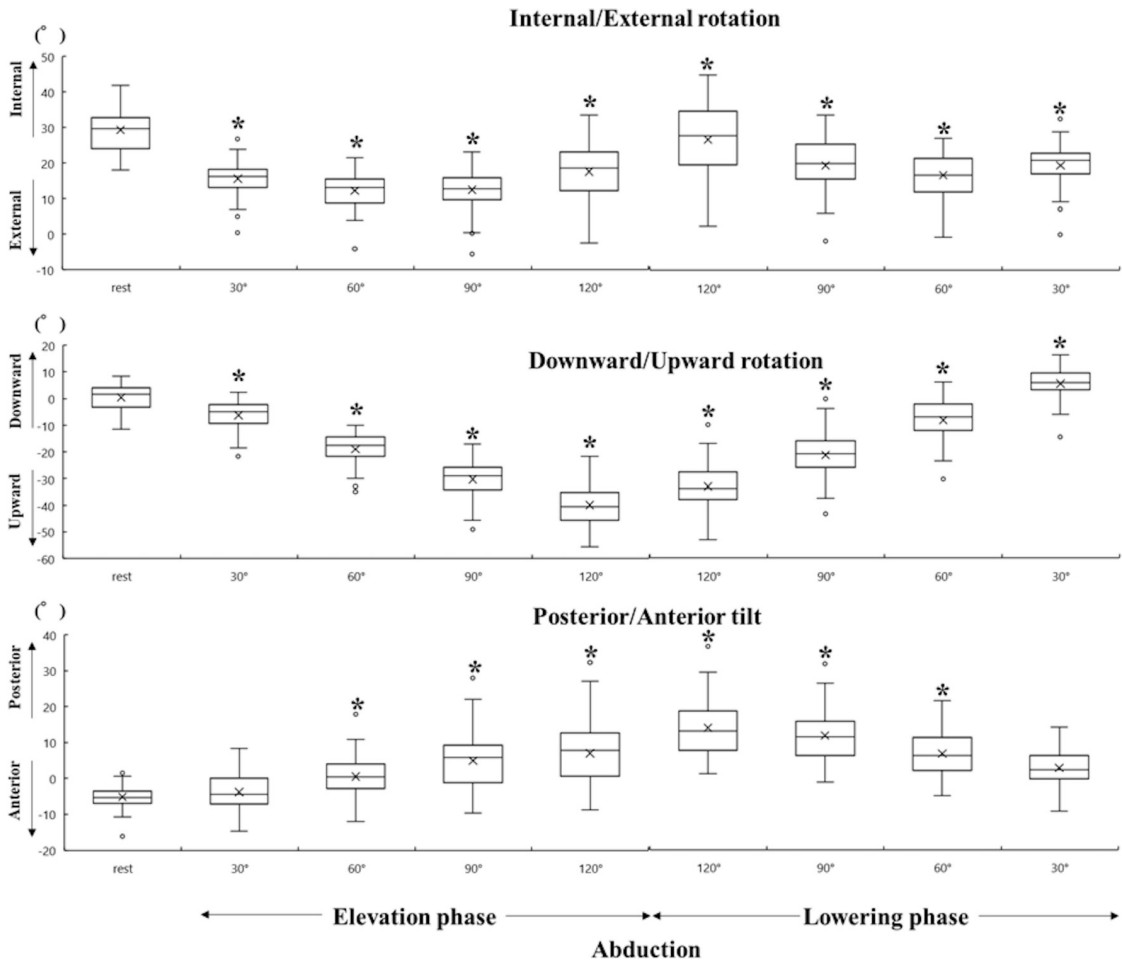

**Fig 4. Scapular position and movement during abduction.** The descriptions are same as the Fig 3.

shoulder movement. Thus, our findings did not only support this previous study but also made the evidence for interaction between scapular initial position and scapular movement during dynamic motions.

The correlations found in the present study could be due to kinematic interaction between humerus and scapula during dynamic motions. For the upward rotation of the scapula, the kinematic characteristic of the humerus and the scapula probably affected the correlations. Inman et al.[24] firstly found the scapulohumeral rhythm, and then many subsequent studies validated the kinematic characteristic between humerus and scapula[8,14,25]. In other words, the angles of the humerus and the scapula constantly shifted through arm elevation. In the present study, through the humeral angles except rest to 30˚, which is called setting phase[26], the integral values to some extent were constant, and the CVs among the participants tended to be small for the upward rotation of scapula. The scapula therefore probably maintains a consistent pattern from rest to the end of movement except setting phase. With respect to internal rotation and posterior tilt no studies have demonstrated a specific pattern between the humerus and the scapula, such as the scapulohumeral rhythm. Our results found that the CVs for internal rotation and posterior tilt tended to be large, indicating the individual variability of scapular movement. On the other hand, compared to that for the upward rotation of the scapula, the integral value for the internal rotation and posterior tilt of the scapula tended to be

**Table 2. Integral value of scapular movement and coefficient of variance.**

| Elevation angle | | | Elevation phase | | | | Lowering phase | | | |
|---|---|---|---|---|---|---|---|---|---|---|
| | | | rest to 30˚ | 30 to 60˚ | 60 to 90˚ | 90 to 120˚ | 120 to 90˚ | 90 to 60˚ | 60 to 30˚ | 30˚ to rest |
| Flexion | Internal/ External Rotation | Integral value (˚) | 3.1 (1.8, 5.9) | 4.9 (4.1, 6.4) | 3.8 (2.2, 5.4) | 3.9 (2.1, 5.6) | 2.7 (1.5, 4.9) | 2.6 (1.6, 3.3) | 3.8 (2.3, 5.8) | 4.6 (2.4, 8.2) |
| | | CV (%) | 83 | 37 | 51 | 64 | 71 | 63 | 54 | 74 |
| | Downward/ Upward Rotation | Integral value (˚) | 5.9 (3.3, 8.0) | 11.6 (10.1, 12.6) | 13.6 (12.2, 15.9) | 9.5 (5.8, 12.0) | 8.5 (6.0, 11.2) | 14.3 (12.9, 15.8) | 12.2 (11.2, 13.8) | 4.6 (1.4, 7.0) |
| | | CV (%) | 53 | 16 | 20 | 42 | 46 | 21 | 18 | 76 |
| | Posterior/ Anterior tilt | Integral value (˚) | 5.8 (3.1, 9.0) | 2.1 (1.6, 3.7) | 2.5 (1.6, 3.6) | 3.4 (2.7, 5.5) | 3.7 (1.6, 6.2) | 2.4 (1.8, 3.4) | 2.4 (1.4, 3.2) | 6.0 (4.1, 10.0) |
| | | CV (%) | 60 | 52 | 55 | 52 | 63 | 57 | 59 | 63 |
| Abduction | Internal/ External Rotation | Integral value (˚) | 23.6 (16.8, 29.8) | 2.7 (1.8, 5.5) | 2.7 (1.6, 4.2) | 5.1 (3.8, 7.7) | 5.4 (4.0, 7.8) | 2.7 (2.1, 4.3) | 2.4 (1.2, 4.4) | 23.3 (19.2, 29.3) |
| | | CV (%) | 38 | 76 | 57 | 44 | 45 | 45 | 76 | 33 |
| | Downward/ Upward Rotation | Integral value (˚) | 34.1 (28.7, 37.5) | 12.6 (11.3, 13.8) | 11.4 (9.6, 13.4) | 9.9 (7.6, 11.9) | 9.9 (7.9, 11.6) | 11.3 (8.3, 13.1) | 11.9 (9.3, 13.1) | 34.2 (29.4, 37.0) |
| | | CV (%) | 19 | 15 | 23 | 29 | 28 | 31 | 25 | 20 |
| | Posterior/ Anterior tilt | Integral value (˚) | 6.4 (3.2, 10.6) | 4.4 (3.2, 5.5) | 4.6 (2.7, 5.9) | 2.7 (1.9, 3.6) | 2.8 (1.9, 3.9) | 5.0 (3.7, 6.7) | 4.1 (3.1, 5.7) | 7.3 (4.5, 13.0) |
| | | CV (%) | 60 | 49 | 54 | 66 | 58 | 48 | 57 | 62 |

Values are expressed as median (25th, 75th). CV, coefficient of variance

small. The scapular movement in these rotations thus would be dependent on the scapular initial position because of the small changes in scapular movement during elevation and lowering, even if the specific characteristic between the humerus and the scapula was not seen.

The present study showed the tendency of the correlation coefficient to decrease with humeral angle for each scapular rotation. Previous studies that investigated the activities of the scapular muscles during elevation and lowering showed that muscle activity around the scapula among participants during arm elevation varied widely with arm angle, although no

**Table 3. Correlation coefficient between scapular initial position and scapular movement.**

| Elevation angle | | Elevation phase | | | | Lowering phase | | | |
|---|---|---|---|---|---|---|---|---|---|
| | | 30˚ | 60˚ | 90˚ | 120˚ | 120˚ | 90˚ | 60˚ | 30˚ |
| Flexion | Internal/External rotation | 0.64** (< .001) | 0.51** (.002) | 0.35* (.041) | 0.29 (.094) | 0.35* (.044) | 0.37* (.034) | 0.37* (.032) | 0.57* (< .001) |
| | Downward/Upward rotation | 0.78** (< .001) | 0.66** (< .001) | 0.57** (< .001) | 0.39* (.022) | 0.39* (.024) | 0.34* (.048) | 0.39* (.019) | 0.55** (< .001) |
| | Posterior/Anterior tilt | 0.77** (< .001) | 0.76** (< .001) | 0.60** (< .001) | 0.43* (.011) | 0.37** (.030) | 0.56** (< .001) | 0.72** (< .001) | 0.76** (< .001) |
| Abduction | Internal/External rotation | 0.61** (< .001) | 0.49** (.003) | 0.37* (.034) | 0.27 (.129) | 0.23 (.201) | 0.26 (.133) | 0.38* (.027) | 0.51** (.002) |
| | Downward/Upward rotation | 0.76** (< .001) | 0.77** (< .001) | 0.69** (< .001) | 0.59** (< .001) | 0.56** (< .001) | 0.64** (< .001) | 0.64** (< .001) | 0.62** (< .001) |
| | Posterior/Anterior tilt | 0.69** (< .001) | 0.53** (< .001) | 0.41* (.015) | 0.40* (.019) | 0.39* (.022) | 0.37* (.030) | 0.47** (.005) | 0.67** (< .001) |

The values are ρ of Spearman's rank correlation coefficient and $P$ values are in parentheses.

*, significant correlation ($P < 0.05$).

**; significant correlation ($P < 0.01$).

statistical analysis was performed[27,28]. Given these previous studies, the correlation coefficients tended to be small with respect to arm angle because the individual variability of scapular movement could be large due to the various activities of scapular muscles.

Three-dimensional measurement of the scapula would require the use of a motion-capture system such as an optical camera or electromagnetic sensors. However, the use of these devices in the clinical setting and on the sports field is not practical with respect to time and cost. In addition, assessment of scapular movement during elevation is difficult and varies among investigators[29]. Our results clarified significant correlations between scapular initial position and scapular movement during flexion and abduction. In other words, it is likely that scapular movement during elevation can be estimated from the scapular initial position. From clinical perspective, although lateral scapular side test and modified one is generally used as diagnostic method to discrete people with or without shoulder dysfunction, some previous studies argued that these test don not have clinical utility because of low accuracy[30,31] and specificity[32]. Baring these clinical situations in our mind, therefore, we advocate that our findings would not be directly applicable for diagnosis of shoulder dysfunction but useful in assessing scapula movement or as criteria for effect of therapeutic exercise because the assessment of the scapular initial position is easy and reliable[33]. We conjecture that deviation of scapular motion at first would be an unneglectable sign of shoulder dysfunction, and then the deviation may alter due to pain and muscle weakness, resulting in the low accuracy and specificity of scapular-based assessments such as the lateral scapular side test. So, prospective study to discover whether the scapular initial position associates with the shoulder dysfunction is of importance.

There are some limitations to the interpretations of our findings. First, all of the participants were healthy young men, as prescribed by the exclusion criteria. Therefore, it is not clear whether the findings can be applied to women, older adults, and individuals with a shoulder disorder such as shoulder instability or rotator cuff tear. Future studies involving these participants are needed. Second, although we analyzed the scapular movement in humeral elevation up to 120˚ in the present study, measurement errors could not been completely excluded because the acromial method always included some errors compared with the percutaneous pinning[23] and the tripod measurement[34]. Therefore, the precise relationship between the scapular initial position and scapular movement without subcutaneous tissue remains unclear. A future study is, therefore, necessary to investigate the precise relationship between the scapular initial position and its movement using 2D/3D registration technique. Moreover, whether this relationship exists with other tasks is unclear because flexion and abduction were the only tasks used for measurement. Thus, other tasks such as the activities of daily living or overhead motion should be the focus of future studies.

## Conclusions

We investigated the relationship between the scapular initial position and scapular movement in elevation and lowering phase during dynamic motions i.e. flexion and abduction in healthy young men. Our results found significant positive correlations in upward and internal rotation and in posterior tilt at all humeral angles, but not for internal rotation at a humeral elevation of 90˚ in lowering phase and 120˚ in both phases. In addition, these correlation coefficients for flexion and abduction along with humeral elevation tended to be small.

## Acknowledgments

The authors thank Satoko Ibuki (Kyoto University) for language editing and proofreading.

## Author Contributions

**Conceptualization:** Jun Umehara, Masahide Yagi.

**Data curation:** Jun Umehara, Tetsuya Hirono.

**Formal analysis:** Jun Umehara.

**Methodology:** Jun Umehara, Masahide Yagi.

**Resources:** Tomohito Komamura.

**Software:** Satoru Nishishita.

**Supervision:** Noriaki Ichihashi.

**Writing – original draft:** Jun Umehara.

**Writing – review & editing:** Masahide Yagi.

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
