## [Decision Letter · Decision Letter 0]

9 Aug 2019

PONE-D-19-19999

Relationship between scapular initial position and scapular movement during dynamic motions

PLOS ONE

Dear Mr Umehara,

Thank you for submitting your manuscript to PLOS ONE. After careful consideration, we feel that it has merit but does not fully meet PLOS ONE’s publication criteria as it currently stands. Therefore, we invite you to submit a revised version of the manuscript that addresses the points raised during the review process.

We would appreciate receiving your revised manuscript by Sep 23 2019 11:59PM. To enhance the reproducibility of your results, we recommend that if applicable you deposit your laboratory protocols in protocols.io, where a protocol can be assigned its own identifier (DOI) such that it can be cited independently in the future. For instructions see: http://journals.plos.org/plosone/s/submission-guidelines#loc-laboratory-protocols

We look forward to receiving your revised manuscript.

Kind regards,

Chunfeng Zhao, MD

Academic Editor

PLOS ONE

Journal Requirements:

2. We note that Figure 1 includes an image of a patient / participant / in the study.

Reviewers' comments:

Reviewer's Responses to Questions

**Comments to the Author**

1. Is the manuscript technically sound, and do the data support the conclusions?

Reviewer #1: Partly

2. Has the statistical analysis been performed appropriately and rigorously? 

Reviewer #1: Yes

3. Have the authors made all data underlying the findings in their manuscript fully available?

Reviewer #1: Yes

4. Is the manuscript presented in an intelligible fashion and written in standard English?

Reviewer #1: Yes

5. Review Comments to the Author

Reviewer #1: The authors investigated the interaction between the scapular initial position and the scapular motions using the motion trackers. Although their findings that the scapular initial positions mostly correlated to the scapular movement were considered important, there are some comments for this study

#1

First, they adopted three motion trackers and secure them to the skin using the tapes. We should carefully consider how accurate they detect the motions. Especially the rotations, the authors demonstrated tiny differences during the motions. It would be helpful for us to understand the outcomes more clearly if the authors could demonstrate the accuracy of the device.

In addition, how did they assess the effect of subcutaneous fat on the outcomes. We can assume the authors selected young males without excessive obesity from the overall data. Please mention these parts more carefully at the Materials and Methods.

#2

They described their hyposthesis for this study: the initial position might correlate to the motions. The major problem of this study was the participants’ characteristics. They only assessed “young” “male” subjects. If the authors would like to prove it, I strongly recommend them to assess the participates including elder people and female people as well.

#3

The authors assessed three times for each motion. Regarding the methodology, please mention how the authors calculate each motions from triplicated data.

In addition, the authors mentioned the reliability of the measurement “the ICC(1,1) calculated from ten healthy men …”. Regarding the reliability, please explain the detail about the values for all motions and rotations, because these information would be helpful to understand the reliability of the methodology using the motion trackers with adhesive tapes.

#4

Regarding my understanding, the measurements using the surface motion trackers have disadvantage for the accuracy even they selected limited motion from 30 to 120 degrees. Some researchers used the percutaneous pinning or fluoroscopic measurement to address to the scapula. I suppose that only the upward/downward rotation of the scapula could be assessed with this measurement tool. I recommend the authors to explain the comparison of various techniques to assess the scapular and/or humerus motions with the accuracy values.

6. PLOS authors have the option to publish the peer review history of their article (what does this mean?). If published, this will include your full peer review and any attached files.

Reviewer #1: No

---

## [Author Response · Author response to Decision Letter 0]

23 Aug 2019

We thank the reviewer for their constructive comments on our manuscript, as well as for the suggestions provided. Our responses to your comments have been uploaded as the file labeled 'Response to Reviewers'. Please find the file.

---

## [Decision Letter · Decision Letter 1]

16 Oct 2019

PONE-D-19-19999R1

Relationship between scapular initial position and scapular movement during dynamic motions

PLOS ONE

Dear Mr Umehara,

Thank you for submitting your manuscript to PLOS ONE. After careful consideration, we feel that it has merit but does not fully meet PLOS ONE’s publication criteria as it currently stands. Therefore, we invite you to submit a revised version of the manuscript that addresses the points raised during the review process.

Although in the revision, you have addressed some concerns from previous review, it is not fully satisfied with editor and reviewer for your response. Please carefully address the questions and concerns that have been raised. Otherwise, your manuscript will be rejected.

We would appreciate receiving your revised manuscript by Nov 30 2019 11:59PM. To enhance the reproducibility of your results, we recommend that if applicable you deposit your laboratory protocols in protocols.io, where a protocol can be assigned its own identifier (DOI) such that it can be cited independently in the future. For instructions see: http://journals.plos.org/plosone/s/submission-guidelines#loc-laboratory-protocols

We look forward to receiving your revised manuscript.

Kind regards,

Chunfeng Zhao, MD

Academic Editor

PLOS ONE

Reviewers' comments:

Reviewer's Responses to Questions

**Comments to the Author**

1. If the authors have adequately addressed your comments raised in a previous round of review and you feel that this manuscript is now acceptable for publication, you may indicate that here to bypass the “Comments to the Author” section, enter your conflict of interest statement in the “Confidential to Editor” section, and submit your "Accept" recommendation.

Reviewer #1: (No Response)

2. Is the manuscript technically sound, and do the data support the conclusions?

Reviewer #1: Partly

3. Has the statistical analysis been performed appropriately and rigorously? 

Reviewer #1: Yes

4. Have the authors made all data underlying the findings in their manuscript fully available?

Reviewer #1: Yes

5. Is the manuscript presented in an intelligible fashion and written in standard English?

Reviewer #1: Yes

6. Review Comments to the Author

Reviewer #1: There are some comments for the revised manuscript.

Comment to Response #1

Page 7 Lines 111-112

Instead of using inches, I think the authors should use the unit of mm, cm, since they have used cm in the next sentence.

Page 5 Lines 79-80

If the authors described the sentence regarding the exclusion criteria for women, they should demonstrate scientific evidence that women have more measurement error because of the subcutaneous fat.

Comment to Response #2

Again, I recommend the authors to include others with additional characteristics such as elder populations, instead of adding the description of “healthy young men”.

Comment to Response #3

The authors modified the description regarding the methodology and its reliability.

Comment to Response #4

They modified the sentence regarding the limitation. I suppose they should have descripted how they justified the internal/external rotation as well as anterior/posterior tilt with scientific values in advance to this study.

7. PLOS authors have the option to publish the peer review history of their article (what does this mean?). If published, this will include your full peer review and any attached files.

Reviewer #1: No

---

## [Author Response · Author response to Decision Letter 1]

21 Oct 2019

Response to Reviewer

Itemized responses to the reviewer’s comments have been listed below in reference to our manuscript entitled “Relationship between scapular initial position and scapular movement during dynamic motions”.

Reviewer #1

We thank the reviewer for their constructive comments on our manuscript, as well as for the suggestions provided.

Our responses to your comments are listed below. For clarity, the parts quoted from the manuscript are in red, and the implemented changes are highlighted in yellow.

Comment #1

Page 7 Lines 111-112

Instead of using inches, I think the authors should use the unit of mm, cm, since they have used cm in the next sentence.

Response

We have changed the unit from inches to mm.

Revised manuscript

The accuracy of the sensor is 0.762 mm …

Comment #2

Page 5 Lines 79-80

If the authors described the sentence regarding the exclusion criteria for women, they should demonstrate scientific evidence that women have more measurement error because of the subcutaneous fat.

Response 

In accordance with your comment, we have cited a previous study investigating sex differences in skeletal muscle, subcutaneous adipose tissue, bone, and so with use of magnetic resonance imaging (Abe et al., 2003). The study included Japanese healthy young men and women as subjects and showed women had higher percentage fat and fat mass than men. This result allows us to assume that women have more measurement error than men because of the subcutaneous fat.

Revised manuscript

…we also excluded women because they would have more subcutaneous fat than men [20], …

Comment #3

Again, I recommend the authors to include others with additional characteristics such as elder populations, instead of adding the description of “healthy young men”.

Response

Thank you for your suggestion. You recommend to include population(s) with additional characteristics such as women and the elderly. However, we beg to disagree that this is necessary. Herein there are two reasons.

1. Populations with additional characteristics such as older and women, to our knowledge, have a possibility to disturb our result, interpretation and conclusion. Previous studies showed the differences in scapular kinematics between asymptomatic older and young population (Saker, 2014) even when the comparison between men and women (Schwartz et al., 2016). These characteristics dependent on the population are due to several factor. For instance, muscle strength loss is inevitable with age and older people have less muscle strength than young (Larsson et al., 1979; Murray et al., 1985). Anthropometric parameter (Winter, 2009), bone shape (Paraskevas et al., 2008), and neuromuscular function (Anders et al., 2004) are different between gender. Namely, additional population in the current study done not make sense.

2. Inclusion of populations such as the elderly and women may create large measurement error between bone and skin due to the subcutaneous tissues. As described in response to your comment#2, women have higher percentage fat and fat mass than men (Abe et al., 2003) and older groups show thicker fat than young groups (Kanehisa et al., 2004).

Bearing in mind these difficulties, we decided to limit the population to healthy young men. As described in our limitations, it is not clear whether the findings can be applied to women, older adults, and individuals with shoulder disorders. However, the lack of inclusion of other populations does not imply that our data do not have scientific value. In the future we plan to conduct these experiment including populations such as older adults and women. In short, we believe that excluding such populations does not make our results invalid.

Since this is the second time you have raised this point, we understand that you feel it is important. However, you have not provided any scientific rationale to convince us of that importance. Moreover, thousands of studies have been published which consider just one population. Population comparisons is therefore clearly not a prerequisite for publication in general. While we agree that a population comparison would be useful, and would also be necessary to demonstrate general validity, we believe that these are separate issues that are tangential to our paper’s message.

If you still feel that it is necessary to add additional populations, we respectfully request that you clarify the scientific rationale for your recommendation.

Comment #4

They modified the sentence regarding the limitation. I suppose they should have descripted how they justified the internal/external rotation as well as anterior/posterior tilt with scientific values in advance to this study.

Response

We apologize, but we do not understand this comment. Please allow us to rephrase your comment as follows: “Authors should use a priori rational to explain why they used internal/external rotation and anterior/posterior tilt.”

If this is what your comment means, we believe that our Introduction and Methods already sufficiently detail the rationale for choosing these tasks.

If this is not what your comment means, we respectfully request clarification.

---

## [Decision Letter · Decision Letter 2]

17 Dec 2019

Relationship between scapular initial position and scapular movement during dynamic motions

PONE-D-19-19999R2

Dear Dr. Umehara,

We are pleased to inform you that your manuscript has been judged scientifically suitable for publication and will be formally accepted for publication once it complies with all outstanding technical requirements.

With kind regards,

Chunfeng Zhao, MD

Academic Editor

PLOS ONE

Additional Editor Comments (optional):

Reviewers' comments:

Reviewer's Responses to Questions

**Comments to the Author**

1. If the authors have adequately addressed your comments raised in a previous round of review and you feel that this manuscript is now acceptable for publication, you may indicate that here to bypass the “Comments to the Author” section, enter your conflict of interest statement in the “Confidential to Editor” section, and submit your "Accept" recommendation.

Reviewer #2: All comments have been addressed

2. Is the manuscript technically sound, and do the data support the conclusions?

Reviewer #2: Yes

3. Has the statistical analysis been performed appropriately and rigorously? 

Reviewer #2: Yes

4. Have the authors made all data underlying the findings in their manuscript fully available?

Reviewer #2: Yes

5. Is the manuscript presented in an intelligible fashion and written in standard English?

Reviewer #2: Yes

6. Review Comments to the Author

Reviewer #2: The authors have addressed the comments raised in the previous review. I think that this manuscript is now acceptable for publication.

7. PLOS authors have the option to publish the peer review history of their article (what does this mean?). If published, this will include your full peer review and any attached files.

Reviewer #2: No

---

## [Editor Report · Acceptance letter]

19 Dec 2019

PONE-D-19-19999R2 

Relationship between scapular initial position and scapular movement during dynamic motions 

Dear Dr. Umehara:

I am pleased to inform you that your manuscript has been deemed suitable for publication in PLOS ONE. Congratulations! Your manuscript is now with our production department. 

With kind regards,

on behalf of

Dr. Chunfeng Zhao 

Academic Editor

PLOS ONE